# Potassium Application Alleviated Negative Effects of Soil Waterlogging Stress on Photosynthesis and Dry Biomass in Cotton

**Li Huang** [1,2], **Jinxiang Li** [1], **Pan Yang** [1,2], **Xianghua Zeng** [1], **Yinyi Chen** [1] and **Haimiao Wang** [1,2,3,*]

[1] Key Laboratory of Ecology of Rare and Endangered Species and Environmental Protection, Guangxi Normal University, Ministry of Education, Guilin 541006, China; huangli8975@126.com (L.H.); 13096695620@163.com (J.L.); tl201130@163.com (P.Y.); xianghua212022@163.com (X.Z.); cyy132022@163.com (Y.C.)

[2] Guangxi Key Laboratory of Landscape Resources Conservation and Sustainable Utilization in Lijiang River Basin, Guangxi Normal University, Guilin 541006, China

[3] College of Life Sciences, Guangxi Normal University, Guilin 541006, China

* Correspondence: wanghaimiao@gxnu.edu.cn

**Abstract:** Soil waterlogging is one of the most serious abiotic stresses on plant growth and crop productivity. In this study, two potassium application levels (0 and 150 kg $K_2O$ $hm^{-2}$) with three types of soil waterlogging treatments (0 d, 3 d and 6 d) were established during cotton flowering and boll-forming stages. The results showed that soil waterlogging markedly reduced RWC (relative water content), gas exchange parameters and cotton biomass. However, potassium application considerably improved the aforementioned parameters. Specifically, 3 d soil waterlogging with potassium increased Pn (net photosynthetic rate), Gs (stomatal conductance), Ci (intercellular $CO_2$ concentration) and Tr (transpiration rate) by 4.55%, 27.27%, 5.74% and 3.82%, respectively, compared with 3 d soil waterlogging under no potassium, while the abscission rate reduced by 2.96%. Additionally, the number of bolls and fruit nodes under 6 d soil waterlogging with potassium increased by 16.17% and 4.38%, compared with 6 d soil waterlogging under no potassium. Therefore, it was concluded that regardless of 3 d or 6 d soil waterlogging, potassium application alleviated the negative effects of waterlogging by regulating the plant water status, photosynthetic capacity and plant growth in cotton. These results are expected to provide theoretical references and practical applications for cotton production to mitigate the damage of soil waterlogging.

**Keywords:** potassium; cotton; photosynthesis; agronomic traits; dry biomass; soil waterlogging

## 1. Introduction

Soil waterlogging is a major agricultural restrictive abiotic stress that constrains plant growth, decreases crop yield and is more frequent because of global climate change [1]. There are three major cotton-producing areas in China, including the northwest inland cotton area, the Yellow River Basin cotton area and the Yangtze River Basin cotton area. Among them, the sown area in the middle and lower reaches of the Yangtze River accounts for about a quarter of the national cotton sown area. However, global climate models predict that rainfall will be distributed more unevenly with global warming [2]. Additionally, the Yangtze River Basin cotton planting area is located in the subtropical monsoon climate and is thus abundant in rain, with an average annual precipitation of about 1200 mm and relatively concentrated. The precipitation from May to October accounts for more than 65% of the annual precipitation [3]. Thus, floods, plum rain and other rainy weather are often encountered and produce waterlogging disasters in the cotton bud period and flower boll period. Moreover, as a result of deficient drainage systems and accelerated land degradation, waterlogging incidents have become increasingly frequent and unpredictable [4]. Simultaneously, cotton growth habits are complex, uncertain and sensitive to

abiotic stress, and are often affected by waterlogging. Waterlogging stress has become one of the main devastating factors limiting crop growth and productivity [1]. Biochemical and physiological processes are more vulnerable to waterlogging stress, especially during the flowering and bolling stages in cotton. The damage to crops caused by waterlogging is mainly due to oxygen deficiency; waterlogging stress leads to the depletion of oxygen in the plant roots, and the rate of oxygen pervasion in waterlogged soil is 10,000 times lower than that in well-drained soil [5]. The growth and development of crops is inhibited in waterlogging stress, which is first reflected in the crop morphology changing significantly. The main manifestations are decreases in terms of the number of leaves, leaf moisture content, biomass accumulation and chlorophyll content [6]. Moreover, cotton plant homeostasis is dislocated, the dynamic balance of reactive oxygen metabolism is imbalanced and finally quality or biomass is significantly reduced under waterlogging stress [7,8].

However, a plant suffering waterlogging stress is able to make certain physiological and molecular adjustments to adapt, but the plant's own regulatory ability is sometimes insufficient, and external forces need to be added to promote the plant's ability to resist adversity [4]. Some of the previously published studies have demonstrated the ability of potassium nitrate modulates, urea, potassium and salicylic acid to resist salinity, flooding and drought [1,9–11]. Nutrient management is one of the efficient ways of mitigating the impact of abiotic stress on plant growth and yield formation development [12], such as potassium fertilizer application [8,10]. Potassium (K) has a significant role in the activation of many enzymes and several vital biological processes, such as stomatal movements, protein formation and energy transformation. Moreover, K can enhance crop tolerance to abiotic stresses [13]. Prospective evidence is available concerning the role of K in activating enzymes regulating hormones, improving gas exchange and optimizing the stomatal conductance and stress resistance in different crop species [8,14]. The application of K has been reported to alleviate the negative effects of waterlogging stress in several crops, including cotton and sugarcane [8,15].

Potassium is an important quality element in cotton, usually constituting 2–10% of plant dry weight, with cotton's critical leaf concentrations occurring at 0.9–1.2% plant dry weight [16], and is highly sensitive to K fertilization. Potassium deficiency affects the antioxidant metabolism and the carbon–nitrogen balance in cotton leaves [17,18]. Additionally, some reports showed that potassium application positively influenced the normal function of stomata, biomass production, biomass partitioning and morphological indices in cotton [19–22]. Comparatively, some scholars have shown that potassium application affects metabolism in the leaf subtending the cotton boll and its relationship with boll biomass, the influence of soil potassium application and foliar potassium application on cotton yield under waterlogging stress and the influence of potassium application on hormone content and endogenous protective enzyme activity in cotton under soil waterlogging stress [8,23,24]. Potassium application improves drought stress alleviation potential in cotton by enhancing photosynthesis, nitrogen metabolism, carbohydrate metabolism and osmotic adjustment [10,25]. Moreover, potassium regulates the metabolism of different endogenous hormones levels to enhance resistance in cotton suffering from waterlogging stress, and promotes cotton growth and development [8]. Therefore, we hypothesized that potassium application could provide a positive effect on cotton subjected to soil waterlogging. Nonetheless, the effect of prior potassium application on photosynthesis and dry biomass accumulation in cotton with subsequent soil waterlogging stress in the flowering and boll-forming stages remains unclear. To address this, this study was conducted to assess the role of potassium in improving the negative effects of soil waterlogging stress on photosynthesis parameters in cotton leaf, and estimate the effects of K application on plant growth and dry biomass accumulation under soil waterlogging stress conditions in the flowering and boll-forming stages. Understanding that how photosynthesis and the growth patterns of cotton plants respond to supplies of K nutrient under waterlogging stress is necessary to determine management practices for improving yields and resource use efficiency.

## 2. Materials and Methods

### 2.1. Experimental Design

The cotton variety used was *Gossypium hirsutum* cv. CCM 45 (growth period 128–135 days) from March to September 2019, and the barrel planting method was used in the Biology Experiment Center of Guangxi Normal University (110°16′ E, 25°05′ N), Yanshan District, Guilin, Guangxi. In this experiment, a randomized block design with two levels of potassium application (K) and three soil moisture (W) treatments was arranged. Thus, the experiment treatments were SW0 (control, no potassium and no soil waterlogging), SW3 (no potassium but 3 d soil waterlogging), SW6 (no potassium but 6 d soil waterlogging), KW0 (potassium application and no soil waterlogging), KW3 (potassium application and 3 d soil waterlogging) and KW6 (potassium application and 6 d soil waterlogging).

Fifty-four pots were established in this experiment. The pots used in the experiment were 38 cm in diameter and 40.5 cm in height (7 gallons per pot). Every two rows of plastic barrels were left between a 40 cm walkway, with row spacing 59 cm and plant spacing 37 cm. Each pot was filled with 25 kg soil; the soil was naturally air-dried and sieved to remove impurities. The soil N (nitrogen), P (phosphorus), K and organic matter contents were 74.8 mg $kg^{-1}$, 18.7 mg $kg^{-1}$, 105.5 mg $kg^{-1}$ and 17.1 g $kg^{-1}$. The data were quoted from another publication by our group [8]. Direct seeding was used in this experiment with three seeds per pot, and then one healthily growing plant was left in each pot by artificial selection at the trifoliate stage. A total of 240 kg $NO^{3-}hm^{-2}$ nitrogen was applied with 40% of basal fertilizer and 60% of boll fertilizer. Phosphorus application was as basal fertilizer at 120 kg $P_2O_5$ $hm^{-2}$. Potassium fertilizer was potassium sulfate for agriculture. Two potassium application rates were set at 0 and 150 kg $K_2O$ $hm^{-2}$ [23], applied at the trifoliate and early flowering stages at 50% application. The relative soil water content $(75 \pm 5)$% was established as a control (SW0), and two soil waterlogging treatments were set to be waterlogged with a 2 cm water layer over the soil for 3 d (SW3) and 6 d (SW6), when 50% of the flowers in the first position of fruiting branches at 6–8 main stem nodes had bloomed.

### 2.2. Leaf Relative Water Content

The sampling time was 9:00 a.m.–10:00 a.m. After handling for 3 d, 6 d, 9 d, 12 d, 15 d and 18 d, the functional leaves of different treated cottons were picked, washed with deionized water, dried with filter paper and the fresh weight (FW) was recorded. Then, the leaves were immersed in a glass dish filled with distilled water for 10 h until the leaves were saturated with water, and then the water on the surface of the cotton leaves was gently dried and the leaf saturation weight (TW) was measured. Afterwards, the sample was put into a kraft paper bag, labeled, deactivated at 105 °C for half an hour, dried at 80 °C to constant weight (DW) and then the relative water content of the leaves was calculated [26]:

$$RWC(\text{relative water content}, \%) = [(FW - DW)/(TW - DW)] \times 100\% \tag{1}$$

### 2.3. Leaf Gas Exchange Parameters

The cotton plants growing uniformly were selected in each treatment on the 10th, 17th, 24th and 31st days after waterlogging treatment, and the net photosynthetic rate (Pn), stomatal conductance (Gs), transpiration rate (Tr) and intercellular $CO_2$ concentration (Ci) of the leaves (the fourth main stem leaves from the top) of the marked parts of cotton plants were measured using LI-6800 photosynthetic apparatus (LI-COR, United States) from 9:00 a.m. to 10:00 a.m. The instrument used an open gas path, and the $CO_2$ concentration was about 400 µmol/mol. The light quantum density (PAR) was set to 1500 µmol/($m^2 \cdot s$) [27].

### 2.4. Agronomic Traits

During the flowering and boll-forming stages period (early July to late August), the marked cotton buds and cotton bolls were measured once every 7 days from the 11 July onwards at 09:00 to 10:00 a.m. Three cotton plants were selected in each treatment, and the number of buds, bolls, fruit branches and the abscission rate of fixed-point cotton plants were measured.

### 2.5. Biomass

The aforementioned two strains of cotton were separated by root, main stem, fruit branch, leaf, bud, flower and boll. The soil was washed and dried. After 30 min of fixation at 105 °C, it was dried to constant weight at 80 °C to determine its dry matter weight.

### 2.6. Statistical Analysis

SPSS 26 statistical software was used for statistical analysis. Post hoc analysis for differences between means was performed by Fisher's least significant difference (LSD) test at 0.05 alpha level. GraphPad Prism 8 software was used for graphing.

## 3. Results

### 3.1. Leaf Water Status Traits

The leaves of the fourth main stem position from the canopy water status traits were substantially affected by soil waterlogging stress and K application. Overall, relative water content showed a downward and then reupward trend with the increase of the number of days since soil waterlogging treatment stopped (Figure 1). Obviously, the RWC substantially was reduced in 3 d and 6 d soil waterlogging stress compared with no soil waterlogging stress; moreover, the RWC of soil treated with potassium was higher than those treated with no potassium. The RWC of KW3 and KW6 was significantly enhanced compared with SW3 and SW6; interestingly, a lesser increase of RWC was observed under severe waterlogging stress (6 d) than a brief soil waterlogging (3 d) with potassium application. The specific performance was as follows: during the waterlogging relief until the sixth day, the soil waterlogging significantly decreased RWC in cotton, especially from the third day to the sixth day. The drop was very large; SW3 treatment decreased by 4.40%, while KW3 decreased only by 0.14%. Moreover, the RWC was decreased more with increases in waterlogged days; the SW6 treatment decreased by 18.5% whereas the KW6 decreased only by 16.20%. As a whole, the RWC gradually increased across most treatments 6 days after the soil waterlogging treatments stopped.

### 3.2. Changes in Leaf Gas Exchange Parameters

Waterlogging stress gradually reduced net photosynthetic rate (Pn) in cotton plants with the days of soil waterlogging stress. At 10 d of stress relief, Pn in the leaves under the simple waterlogging groups was decreased significantly with the number of days of waterlogging compared with normal control irrigation, and the same trend was observed at 17 d, 24 d and 31 d (Figure 2A). At 10 d after soil waterlogging relief, the Pn of KW0, KW3 and KW6 in the potassium-applied groups increased by 3.40%, 5.47% and 4.87% compared to the waterlogged groups of SW0, SW3 and SW6, respectively. The Pn of KW0, KW3 and KW6 treatments were increased by 3.19%, 3.94% and 4.52% compared to no-potassium in the simple waterlogging stress groups at 17 d. Increases of 3.24%, 4.37% and 4.07% in the Pn of KW0, KW3 and KW6 treatments were found at 24 d. Finally, at 31 d of stress relief, the potassium-applied groups showed increases of 2.52%, 4.46% and 4.73%, respectively, compared to SW0, SW3 and SW6. Pn showed the highest level of increase under potassium application groups which, respectively, increased by 3.40%, 5.47% and 4.87% at 10 d of soil waterlogging treatment. It was found that the longer the waterlogging stress was prolonged, the more difficult the damage to the cotton was to alleviate.

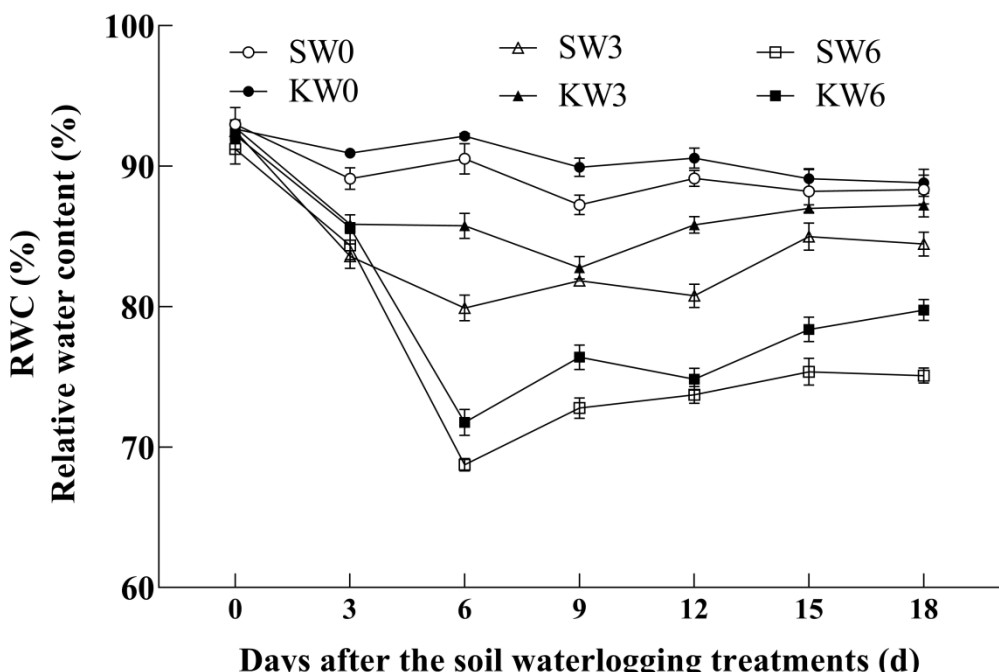

**Figure 1.** Effect of potassium application on leaf relative water content in cotton under soil waterlogging stress. The experiment treatments were SW0 (no potassium application and no soil waterlogging), SW3 (no potassium application and 3 d soil waterlogging), SW6 (no potassium application and 6 d soil waterlogging), KW0 (potassium application and no soil waterlogging), KW3 (potassium application and 3 d soil waterlogging) and KW6 (potassium application and 6 d soil waterlogging).

The Gs of potassium treatment was increased compared to without potassium treatment on the whole (Figure 2B). Specifically, at 10 d after soil waterlogging stress disappeared, Gs showed a trend of decreasing and then increasing with the increase of soil waterlogging days. Gs of potassium treatment was lower than without potassium treatment, but the difference between the two was not big. Interestingly, the difference between the potassium treatment and no-potassium treatment was increased from 17 d. Gs in KW0 and KW6 treatments increased by 109.93% and 29.95% compared with SW0 and SW6 at 17 d after the waterlogging stress was stopped. Later on, the Gs in KW0 and KW3 treatments were increased by 20.52% and 114.25% after 24 d of waterlogging relief. At 31 d after stress was stopped, the Gs of KW0 and KW3 treatments was increased by 89.27% and 19.72%, respectively.

In this study, the Ci of potassium treatment was greater than without potassium on the whole (Figure 2C). The Ci of KW0 and KW3 was increased by 9.58% and 5.74% compared to SW0 and KW3, whereas the Ci in KW6 was decreased by 0.39% compared to SW6. To be specific, on the 10th day, the Ci of the KW0 and KW6 treatments was decreased compared to SW0 and SW6, but the Ci of the KW3 treatment was increased by 2.81% relative to SW3. On the 17th day, Ci increases of 28.47% and 4.43% in the KW0 and KW6 treatments were found compared with SW0 and SW6, respectively. At this time, no difference was found for the Ci of KW3 compared with SW3. On the 24th day, regardless of waterlogging or not, the Ci treated with potassium was higher than that without potassium, increasing by 4.48%, 17.04% and 0.37%, respectively. Additionally, on the 31st day, the Ci of K and KW3 was increased by 7.12% and 3.64%, respectively. Conversely, the Ci of KW6 decreased by 4.38%.

Overall, Tr increased with the length of time that soil waterlogging stress was stopped (Figure 2D). On the 10th day, the variation tendency of Tr and Gs were similar in each treatment; however, the differences between KW0 and SW0 and KW6 and SW6 were quite large. The Tr in the KW0 and KW6 treatments was increased by 28.41% and 17.48%, respectively, compared with SW0 and SW6, while the KW3 treatment was merely increased

by 5.95%. On the 17th day, the Tr in KW0 and KW6 treatments were increased by 18.12% and 5.18%, respectively, compared with SW0 and SW6, while the KW3 treatment was decreased by 11.37%. On the 24th day, the KW3 and KW6 treatments were increased by 28.16% and 4.78%, respectively, compared with SW3 and SW6, while KW0 treatment was decreased by 12.56% compared with SW0. On the 31st day, there was little difference between the potassium treatment and no potassium treatment, but KW6 treatment was decreased by 2.93% compared with SW6.

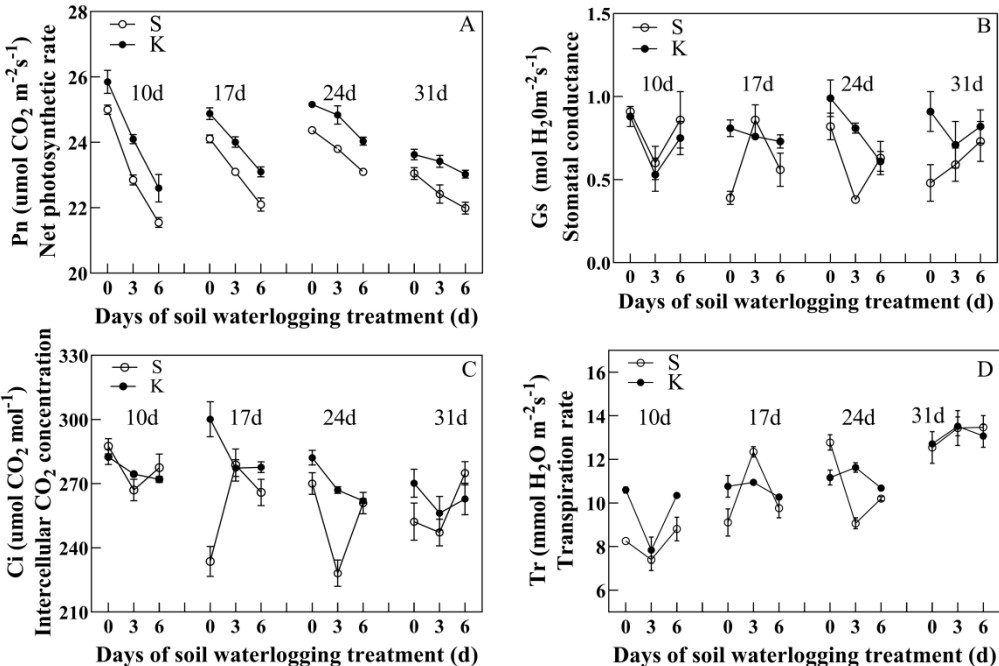

**Figure 2.** Effect of potassium application on gas exchange parameters ((**A**) net photosynthetic rate; (**B**) stomatal conductance; (**C**) intercellular $CO_2$ concentration; (**D**) transpiration rate) in cotton leaves under soil waterlogging stress. S represents the soil waterlogging without potassium treatment groups. K represents the soil waterlogging with potassium treatment groups; 0, 3, 6 in X−coordinate represent the duration days of soil waterlogging treatment and 10 d, 17 d, 24 d, 31 d represent the gas exchange measurement days after soil waterlogging.

### 3.3. Agronomic Traits

As shown in the figure, the trend of the number in cotton buds in each treatment was similar, showing the trend of increasing first, then decreasing, and finally increasing slowly; however, the latter increase was not large, and did not increase to the initial amount (Figure 3A). The longer the duration of the soil waterlogging period, the greater the reduction of the number of buds in cotton. In addition, the number of buds in the SW0 and KW0 treatments was increased on the 25 July, then began to decline. However, the number of buds in both 3 days and 6 days soil waterlogging treatments were increased on the 18 July; in other words, waterlogging may cause cotton to end the production of buds early, and the number of buds in cotton did not reach the number of non-waterlogging treatment. Additionally, the date of the number of buds increasing without potassium treatment was 22 August, but the date of the number of buds increasing in the KW3 and KW6 treatments was 15 August, indicating that the application of K fertilizer can increase the number of buds even earlier.

From the above figures, we found that the trend of number of bolls, fruit nodes and the abscission rate was similar, showing a rapid increasing trend and then a slowly slowing trend with the change of time (Figure 3B–D). In contrast, bolls and fruit nodes showed similar trends between the W0 and W3 treatments, while bolls and fruit nodes in the W6 treatment showed differences in the presence or absence of potassium application. The

number of cotton bolls and fruit nodes were all different on 8 August, but the difference between the potassium application and no potassium application was not large. The number of bolls treated by SW6 and KW6 showed a difference from 18 July, and the difference gradually increased to 1 August, the maximum increase reached 31.09%, and then gradually decreased. The number of fruit nodes also showed the same trend; the maximum increase in the number of fruit nodes reached 10.95%.

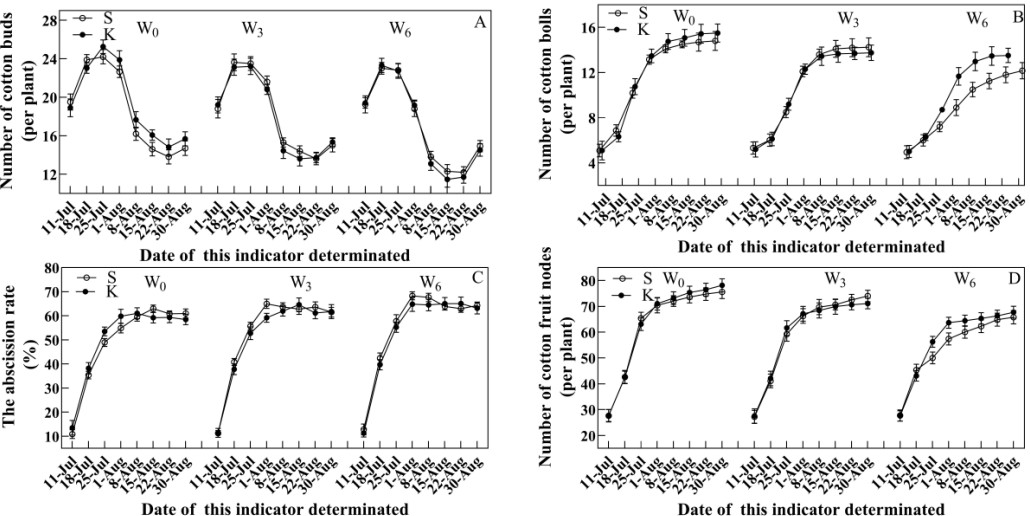

**Figure 3.** Effect of potassium application on agronomic traits ((**A**) number of cotton buds; (**B**) number of cotton bolls; (**C**) the abscission rate; (**D**) number of cotton fruit nodes) in cotton under soil waterlogging stress. S represents the soil waterlogging without potassium treatment groups. K represents the soil waterlogging with potassium treatment groups. W0, W3 and W6 represent 0 d, 3 d and 6 d soil waterlogging treatment, respectively.

*3.4. Dry Matter Accumulation and Allocation*

As shown in Table 1, soil waterlogging stress applied during the flowering stage reduced the photo-assimilate accumulation and partitioning into the plant organs of cotton (Table 1). In particular, the biomass of root and bud flower boll was significantly reduced under soil waterlogging stress. Except for the KW3 treatment, the biomass of the main stem, fruit branch and leaf were decreased with increasing waterlogged days with or without potassium application, but there was no significant difference. Notably, the biomass of root and bud flower boll was significantly reduced with increasing waterlogged days without potassium application; the biomass of root in the SW3 and SW6 treatments declined by 35.79% and 52.32%, respectively. Meanwhile, the biomass of bud flower boll in the SW3 and SW6 treatments was reduced by 17.64% and 45.29%, respectively. However, under potassium application, root biomass was increased by 6.79%, 13.00% and 11.23% in the KW0, KW3 and KW6 treatments, respectively. Additionally, the biomass of bud flower boll was increased by 8.15% and 4.51%, respectively, in KW0 and KW6 compared with SW0 and SW6; remarkably, the biomass of bud flower boll in the KW3 treatment was reduced by 16.63% compared with SW3.

Dry matter accumulation is the basis of cotton yield, and the formation process can be optimized by studying the distribution of dry matter in various organs. In biomass allocation, bud-boll allocation from each treatment was dominant, followed by leaf biomass and main stem biomass, again for fruit branch biomass, and finally for root allocation (Figure 4). The bud–boll allocation in SW0, SW3, SW6, KW0, KW3 and KW6 accounted for 42.58%, 35.65%, 33.45%, 44.40%, 40.32% and 32.38%, respectively; leaf biomass accounted for 22.14%, 24.86%, 25.45%, 21.05%, 22.95% and 26.76%, respectively; the main stem biomass occupied 17.26%, 20.80%, 22.38%, 16.02%, 19.96% and 22.85%, respectively; branches accounted for 11.41%, 12.84%, 13.91%, 11.72%, 11.89% and 13.64%, respectively; and finally, the allocation of roots accounted for 6.61%, 5.85%, 4.82%, 6.81%, 4.88% and 4.38%.

Compared with SW0, the proportion of the root biomass and the bud flower bolls biomass in the SW3 and SW6 treatments was decreased by 11.53%, 27.18%, 16.28% and 6.18%, respectively; conversely, the main stem, leaf and branch allocation were all increased. Moreover, the biomass allocation of root main stem, leaves and branches in the KW3 and KW6 treatments was decreased compared with the SW3 and SW6 treatments. Interestingly, the biomass allocation of bud flower bolls was decreased as waterlogging time increased. The specific performance is as follows: the biomass allocation of bud flower bolls in KW0 treatment was increased by 4.27% compared with the SW0 treatment, and the biomass allocation of bud flower bolls in the KW3 treatment was increased by 13.10% compared with the SW3 treatment, but the biomass allocation of bud flower bolls in the KW6 treatment was decreased by 3.20% compared with SW6.

**Table 1.** Effect of potassium application on plant biomass in waterlogged cotton (g plant$^{-1}$).

| Experimental Treatments | Root | Main Stem | Fruiting Branch | Leaf | Bud Flower Bolls | Total Biomass |
|---|---|---|---|---|---|---|
| SW0 | 23.72 ± 0.64 b | 61.88 ± 3.74 a | 79.38 ± 1.51 a | 40.93 ± 2.41 b | 152.69 ± 3.68 b | 358.58 ± 4.14 b |
| SW3 | 15.23 ± 1.32 d | 62.26 ± 4.90 a | 71.57 ± 6.00 c | 37.10 ± 1.05 c | 125.75 ± 5.69 c | 311.90 ± 4.86 cd |
| SW6 | 11.31 ± 0.35 e | 58.95 ± 0.48 b | 69.04 ± 0.86 d | 35.18 ± 6.52 e | 83.54 ± 1.73 d | 258.01 ± 7.52 e |
| KW0 | 25.33 ± 1.35 a | 59.60 ± 5.55 a | 78.30 ± 5.02 a | 43.59 ± 1.55 a | 165.14 ± 10.77 a | 371.94 ± 0.00 a |
| KW3 | 17.21 ± 0.88 c | 61.17 ± 3.83 a | 73.12 ± 5.59 b | 37.75 ± 1.32 c | 104.84 ± 2.67 c | 294.09 ± 6.31 d |
| KW6 | 12.58 ± 0.66 e | 58.43 ± 0.77 b | 66.43 ± 8.82 d | 36.31 ± 5.89 d | 87.31 ± 5.37 d | 261.05 ± 3.87 e |

Note: Different lowercase letters in the same column indicate significant differences between treatments ($p < 0.05$). The experiment treatments were SW0 (no potassium application and no soil waterlogging), SW3 (no potassium application and 3 d soil waterlogging), SW6 (no potassium application and 6 d soil waterlogging), KW0 (potassium application and no soil waterlogging), KW3 (potassium application and 3 d soil waterlogging) and KW6 (potassium application and 6 d soil waterlogging).

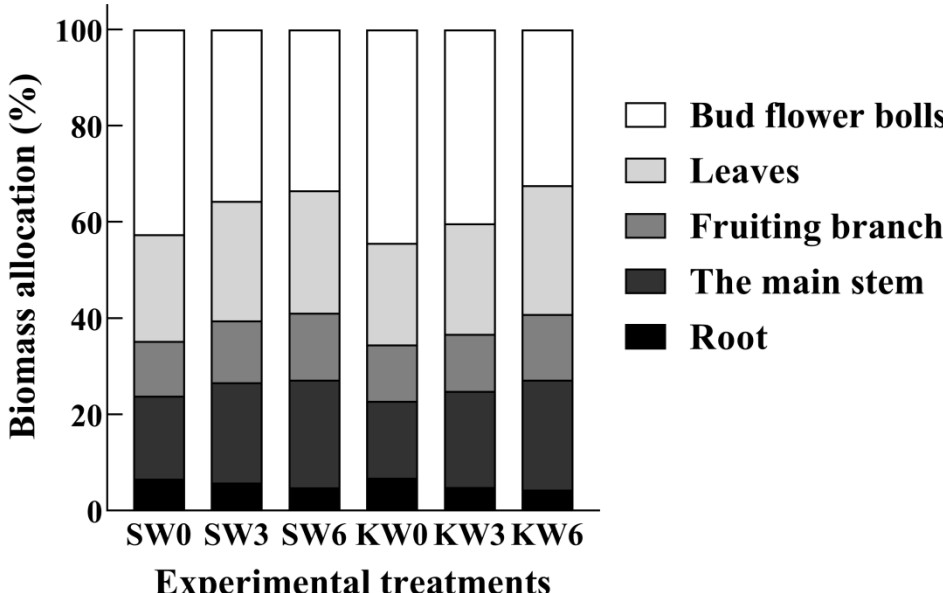

**Figure 4.** Effect of potassium application on biomass allocation in cotton under soil waterlogging stress. The experiment treatments were SW0 (no potassium application and no soil waterlogging), SW3 (no potassium application and 3 d soil waterlogging), SW6 (no potassium application and 6 d soil waterlogging), KW0 (potassium application and no soil waterlogging), KW3 (potassium application and 3 d soil waterlogging) and KW6 (potassium application and 6 d soil waterlogging).

## 4. Discussion

Cotton subjected to soil waterlogging stress can be self-regulated through escape mechanisms, stationary adaptation mechanisms and self-regulatory compensation mechanisms [4,28]. In view of the results confirmed here, it is quite clear that soil waterlog-

ging stress has an inhibitory effect on the production formation process of cotton plants. However, K fertilization of an appropriate dose may not only improve tolerance against waterlogging stress, but also can enhance crop growth and productivity in virtue of proper nutrition. In this study, the application of K fertilizer with 150 kg $K_2O$ $hm^{-2}$ increased the potential of cotton plants to attenuate the negative effects of follow-up waterlogging stress on leaf water relations, photosynthetic capacity and dry biomass accumulation.

### 4.1. Potassium Application Improves Leaf Relative Water Content and Photosynthetic Capacity in Cotton Subjected to Soil Waterlogging Stress

We observed in the present study that the cotton potential to attenuate and recoup the destructive effects of waterlogging stress was enhanced by potassium application. Waterlogging sensitivity is closely related to photosynthetic inhibition in cotton [29]. Photosynthesis is a major source of plant growth and dry matter accumulation, and leaves play a core role in plant energy capture and carbon assimilation process with the potential ability to adapt to climatic and environmental stresses, and thus, they can constantly adapt and develop functions to respond to stress conditions [30]. When plants encounter waterlogging stress, the normal aerobic respiration is blocked, which then directly affects other physiological indicators of plants. Our results agree with previous studies which revealed that soil waterlogging stress could indeed reduce the relative water content of cotton leaves, and potassium application could improve the normal leaf growth index to some extent [25]. We observed that the trend of relative water content in cotton leaves at the fourth main stem position from the canopy showed a decline on the third day after the soil waterlogging treatments stopped and then rose. On the 18th day after waterlogging stopped, the relative water content of 3 d waterlogging stress was basically insignificant between potassium application and without potassium, and it was higher than that of the third day after waterlogging, which indicates that the cotton was free from the effects of short-term waterlogging stress. However, the difference of relative water content between potassium application and without potassium under 6 d soil waterlogging was relatively large, indicating that the longer the duration of waterlogging time, the more severe the negative effects on the plant leaves, and it also takes longer to recover from the waterlogging damage to the cotton, which was consistent with the previous study [8].

The decline of photosynthesis is a common physiological response of cotton to waterlogging stress and also the cause of cotton production [31]. The effect of soil waterlogging stress on cotton photosynthesis is directly reflected by the gas exchange parameters. Net photosynthetic rate (Pn), transpiration rate (Tr) and stomatal conductance (Gs) are the main parameters reflecting the photosynthetic capacity of plants [32]. In this study, we observed that the gas exchange parameters of the functional cotton leaves were affected by soil waterlogging stress. Relative water content and transpiration rate are important plant water relation parameters that are influenced by waterlogging stress. Plants adapt to waterlogging stress with declined stomatal conductance [33]. Changes in stomatal conductance are similar to the intercellular $CO_2$ concentration (Ci), and $CO_2$ will have more difficulty entering the plant leaves as the stomatal void decreases [34]. This study showed a regular decline in Pn (Figure 2A). Since stomatal closure results in a drastic reduction of intercellular $CO_2$ concentration, this might have increased mesophyll resistance in waterlogging-stressed plants.

K is crucial for optimum function of the photosynthetic organ of the plant [10]. In this study, Gs, Tr, Pn and Ci treated with potassium were significantly improved compared to without potassium, indicating that the adverse effect of soil waterlogging stress on photosynthesis in cotton could be effectively alleviated to some extent. Stomata are sensitive to environmental and physiological changes and play an important role in regulating leaf senescence [35]. Additionally, the application of K was proved to be effective in regulating the opening and closing of stomata associated with transpiration rate [36]. Improved stomatal conductance was recorded in waterlogging-stressed plants under K application, which also indicates the role of K in leaf moisture retention and its uptake from soil. K

plays an significant role in stomatal regulation [37], and an inducing role to enhance photosynthesis, maintain water relation components and absorb nutrients; this may have been the reason for the increased photosynthetic parameters and water relation content as observed in this study.

### 4.2. Potassium Application Developed Better Agronomic Traits in Cotton Suffering from Soil Waterlogging Stress

Potassium application has the potential to increase the yield of cotton affected by soil waterlogging stress. Previous studies have pointed out that potassium application increased the weight and the number of bolls per plant in cotton [38]. Cotton lint yield is dependent on boll density, lint percentage and boll weight [39]. The differences of yield components and their contribution to yield decrease may be due to the changes of waterlogging duration and the growth stage at which the stress was imposed [40–42]. Additionally, crop yield loss was detected under more frequent extreme climatic events, especially for extreme waterlogging stress [43]. In this study, waterlogging decreased the number of buds, bolls and fruit nodes, and the decrease for 6 d soil waterlogging was far more than the group which underwent a brief soil waterlogging (3 d). This result was in agreement with Zhang et al. that yield loss increases with waterlogging duration [41,44]. Potassium application increased the number of buds, bolls and fruit knots, and reduced the shedding rate of cotton bolls compared to the cotton without potassium in our study; this result was consistent with previously published studies that potassium application increases the number of bolls per plant, seed and lint yield in cotton [44]. Consequently, the lower shedding rate under K application may be due to an increased number of bolls as a result of compensation. In addition, some studies have reported that the reduction of cotton yield under soil waterlogging stress is ascribed to the reduced number of bolls rather than to reduced lint percentage or boll weight [40,44]. According to the result, we also found that the difference between KW0 and SW0, and KW3 and SW3, on the number of bolls was basically not obvious, but the difference between KW6 and SW6 on the number of bolls was significant, which indicates that potassium application can alleviate negative effects of severe soil waterlogging on cotton boll number; the number of cotton fruit nodes also showed such cases. Therefore, the negative impact of soil waterlogging stress on cotton can be alleviated by potassium application.

Furthermore, an increased shedding rate under soil waterlogging stress conditions greatly accelerated the reduction of the boll number [40]. In the experiment, potassium application reduced the shedding rate of cotton compared to no potassium application. In the absence of soil waterlogging stress, the application of potassium reduced the shedding rate for a few days; however, under soil waterlogging stress, the effect of potassium application was shown from the beginning, which may be explained as soil waterlogging stress being faster and more actively stimulating the cotton protection system by the regulatory mechanisms of triggering endogenous hormones and protective enzymes balance [8]. Therefore, legitimate applications of potassium fertilizer in cotton can increase the number of buds and fruit nodes and reduce the abscission rate of cotton buds, which finally enhances cotton production.

### 4.3. Effect of Potassium Application on Dry Matter Accumulation and Distribution in Cotton Undergoing Soil Waterlogging Stress

Stress closes the stomata of the leaves, which thus affects the Calvin cycle of photosynthesis and causes plants to change from aerobic respiration to anaerobic respiration, reduces energy utilization efficiency [45] and inhibits other biosynthetic activities, which directly leads to the weakening of plant metabolic activities and ultimately reduced accumulation of organic matter under waterlogging stress [4]. Therefore, plant dry weight is an important indicator to evaluate the growth status under soil waterlogging stress. In this study, biomass of leaf, fruiting branch, root, bud flower boll and the whole plant were significantly reduced by waterlogging. The cause of decrease in root biomass was presumably due to decaying of roots, and the decreased vegetative growth possibly resulted

from waterlogging restricting the supply of nitrogen to the shoot, which was observed in a previous experiment by Trought and Drew [46], probably due to waterlogging also constraining the nitrogen nutrients in the soil. However, the biomass of the root was increased under potassium application circumstance, which might be due to allocation of more carbohydrate to the underground portion as the cotton developed adventitious roots to overcome soil waterlogging stress [4]. Moreover, the reduction in the reproductive biomass was primarily due to an increasing shedding rate caused by the waterlogging, which was also shown in this study, and Pn was decreased, which ultimately decreased photo-assimilate synthesis and distribution [47]. The limited carbohydrates resulting from the reduction in Pn also led to the reduced vegetative and reproductive biomass [28].

Potassium application can facilitate stomatal regulation; the stomatal regulation under optimum K application conditions is considered significant for promoting photosynthetic rates, transport of photosynthate to reservoirs and roots, resulting in increasing dry matter production [48]. In addition, potassium can greatly enhance the fixation of $CO_2$ in photosynthesis, which facilitates the distribution and utilization of photosynthetic products [49]. In this study, potassium application significantly increased the plant biomass under waterlogging stress as compared to plants without potassium application (Table 1), due to the positive effect of potassium application on the accumulation of biomass under stress conditions [21]. In contrast, the biomass of the vegetative and reproductive organs of plants with K application was significantly increased under waterlogged conditions compared with the cotton of the no-potassium application. The results of the current experiment also showed that the biomass of the stem was increased, which might be attributed to the fact that the supply of K facilitate promoted to shift dry matter from leaf to stem, enhancing the translocation of more assimilation substances from source to sink [50]. An adequate K supply is known to play a crucial role in phloem translocation of assimilation substances [51]. In no-waterlogging treatments, the biomass of leaves and buds of potassium application was increased compared with no K supply. The increased of leaf and bud flower boll biomass possibly resulted from K transport from older to younger plant tissues, which ensured redistribution of this ion toward growing tissues, such as developing leaves and fruits, as observed in a previous experiment by Mengel and Kirkby [50].

## 5. Conclusions

In this study, we concluded that pre-application of K with 150 kg $K_2O$ hm$^{-2}$ enhanced tolerance against soil waterlogging stress in the cotton flowering and boll-forming stages, especially for brief soil waterlogging stress, primarily indicated by enhanced relative water content in cotton leaves at the fourth main stem position from the canopy, photosynthetic rate and improved agronomic traits. Specifically, 3 d soil waterlogging with potassium application increased Pn, Gs, Ci and Tr by 4.55%, 27.27%, 5.74% and 3.82% compared with the treatment of 3 d soil waterlogging but no potassium application; by comparison, the abscission rate was reduced by 2.96%. Additionally, the number of bolls and fruit nodes in the treatment of 6 d soil waterlogging with potassium application increased by 16.17% and 4.38%, respectively, compared with 6 d soil waterlogging but no potassium application treatment. Overall, regardless of 3 d soil waterlogging or 6 d soil waterlogging, the impact of soil waterlogging stress reduced by the prior potassium application and consequently photosynthetic physiological parameters improved, as a potential mitigating strategy for enhancing the growth of cotton and alleviating the adverse effects of soil waterlogging stress. This study provides a theoretical reference for understanding the roles of nutrient elements in improving plant resistance to environment stress, and has practical applications for cotton production in areas affected by soil waterlogging.

**Author Contributions:** L.H. and H.W. wrote the paper. P.Y., J.L., X.Z. and Y.C. drew the figures and tables. All the authors contributed to the manuscript's editing and revision. All authors have read and agreed to the published version of the manuscript.

**Funding:** We appreciate financial support from the Natural Science Foundation of Guangxi (2018GXNSFBA050036), Sustainable Development Innovation and Key Program of Guangxi Normal University (2018ZD005, 2020CX004). We are also grateful for the Research Funds of the Guangxi Key Laboratory of Landscape Resources Conservation and Sustainable Utilization in Lijiang River Basin (LRCSU21Z0317) and Key Laboratory of Ecology of Rare and Endangered Species and Environmental Protection (Guangxi Normal University), Ministry of Education, China.

**Institutional Review Board Statement:** Not applicable.

**Informed Consent Statement:** Not applicable.

**Data Availability Statement:** Not applicable.

**Conflicts of Interest:** The authors declare no conflict of interest.

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
