# Peer review of "Potassium Application Alleviated Negative Effects of Soil Waterlogging Stress on Photosynthesis and Dry Biomass in Cotton"

_agronomy, doi:10.3390/agronomy13041157_

Round 1

Reviewer 1 Report

Manuscript Number: agronomy-2340420

Title:  Potassium application alleviated negative effects of soil waterlogging stress on photosynthesis and dry biomass in cotton

General comments:   In this study, two potassium application levels (0 and 150 kg K2O hm−2) with three types of soil waterlogging treatments (0d, 3d and 6d) were established during cotton flowering and boll forming stages. The results showed that the waterlogging treatment markedly reduced RWC, gas exchange parameters and dry biomass in cotton plants. However, potassium application considerably improved the aforementioned parameters. Specifically, KW3 on Pn, Gs, Ci, Tr increased by 4.55%, 27.27%, 5.74% and 3.82% compared with the treatment of SW3 respectively, while the abscission rate reduced by 2.96%. Besides, the number of bolls and fruit nodes in KW6 increased by 16.17% and 4.38%, compared with SW6. Furthermore, the biomass of root, fruiting branch, leaves, bud flower bolls was increased in KW3 and KW6, conversely, the biomass of the main stem was reduced, relative to SW3 and SW6 separately. Therefore, it was concluded that prior potassium application can resist the negative effects of waterlogging and regulate the plant water status, photosynthetic capacity, and the plant growth in cotton subjected to waterlogging stress. These results were expected to provide theoretical reference and practical applications on cotton production to mitigate the damage of soil waterlogging.

The study is meeting the scientific standard in terms of technical language, experimental design and implication of the results. The English language of article seems better, but some typo errors should be checked briefly during revision. I suggest minor revision for this article and will be happy to review its revised version. The following suggestions should be considered to improve the manuscript quality before its publication in Agronomy.

Specific comments:

1-     Please re-check the abbreviations which have been mentioned in the whole manuscript. Please elaborate them at least once at its first place in the abstract and other sections of manuscript.

2- In the abstract, authors are asked to state the conclusion of their study.

3-     The figures in the results section are of good quality, but sometimes the fonts are so small that they are hard to be read.

4-     The introduction section is well written and in my opinion it sheds light on the problem in a concise manner. Overall, the objectives are briefly explained and are good to go with.

5-     Materials and methods section is nicely presented and well described. Some references are missing.

6- Please be checked the whole manuscript, the format should be improved such as there are some unit presentations, put the relevant unit with each value across the manuscript.

7- Please add some recent literature to discuss your results in the discussion section.

8- Please re-write the conclusion. The conclusion is recommended to be supported by the data shown in tables, put detail of any limitations of this study, describe implications of this study and provide recommendations for future perspectives.

Minor editing of English language required (some typo errors and grammar errors)

Author Response

We would first like to thank the editor and independent reviewer for their review of our manuscript. We feel that the comments were important for the improvement of our paper.

In response to comments from Reviewer 1:

  1. We have re-checked the abbreviations all through the whole manuscript and elaborate them at its first place in the abstract and other sections of manuscript. Thank you for your suggestion.
  2. In the abstract, we have included conclusion statement. Thank you for noticing.
  3. Thank you for your approve. We have increased the fonts in all figures. Thank you for noticing.
  4. Thank you for your approve.
  5. Thank you for your approve. We have included the references in Materials and methods.
  6. We have double checked the format especially for the unit presentation through the whole manuscript. Thank you for noticing.
  7. We have included some recent literature in the discussion section and revised carefully, as requested. Thank you for your suggestion.
  8. We have re-writed the conclusion, as requested. Thank you for your suggestion.

Reviewer 2 Report

General comment:

The authors raise an important problem in cotton cultivation, i.e. waterlogging stress. Potassium treatment can contribute to plant protection and regulate plant hydration as well as photosynthesis and ultimately cotton growth. The presented study is interesting not only from the scientific point of view but also from the application point of view for cotton production.

Detailed notes are provided below.

Abstract: 

The abstract is well structured but too long The abstract should contain a maximum of about 200 words; 

Introduction: 

Please present the aim of the work, and propose a hypothesis or research problem. The authors should place this point at the end of the introduction, referring to the previously outlined mechanisms that are activated in cotton during drought and to the role of potassium in protecting cotton from stress. 

Results: 

Please improve the readability of figures and tables. For example, figure 3 is too poor quality, table 1 is illegible.

Discussionand Conclusions

Discussion and conclusions written in great detail and exhaustively. Only in the applications, I ask the authors to change the abbreviations of individual treatments to full names, specifying, for example, the number of K doses used (lines 471-474). This should make the conclusions clearer.

References: 

References have small errors. For examples of bold, italics should be corrected and checked all the citations and references according to journal instructions and guidelines.

Author Response

We would first like to thank the editor and independent reviewer for their review of our manuscript. We feel that the comments were important for the improvement of our paper.

In response to comments from Reviewer 2:

Thank you for your approve. We agree with all comments from the reviewer and have revised carefully, as requested.

Abstract:

We have revised the Abstract to contain less 200 words, as requested. Thank you for your suggestion.

Introduction:

We have proposed a hypothesis and included the aim of this study at the end of the introduction, as requested. Thank you for your suggestion.

Results:

We have improved the readability of figures and tables. Figures' dpi changed to 600, fonts in figures were all increased. Table format was editable. Thank you for noticing.

Discussion and Conclusions

Thank you for your approve. We have changed the abbreviations of individual treatments to full names and added the number of K doses in Conclusions. Thank you for your suggestion.

Reviewer 3 Report

This study concludes that prior potassium application can help cotton plants resist the negative effects of waterlogging stress and improve plant growth. The findings have practical implications for cotton production in areas affected by soil waterlogging. Some concerns below need to be addressed before publication.

Line 21, The full name of RWC should be provided, Leaf relative water content?

Line 35-36. The cited ref Goyal et al., does not support this statement. Please consider citing the Refs below:

Liu, K., Harrison, M.T., Yan, H. et al. Silver lining to a climate crisis in multiple prospects for alleviating crop waterlogging under future climates. Nat Commun 14, 765 (2023).

Line 46-47. Could cite refs below

Yan, et al., 2022. Crop traits enabling yield gains under more frequent extreme climatic events. Science of the Total Environment, 808, 152170.

Liu et al., 2022. Designing high-yielding wheat crops under late sowing: a case study in southern China. Agronomy for Sustainable Development, 42(2), 29.

Fig. 1 Remove (100%) from Y axis. Should be RWC(%); delete stopped on X axis

Line 189-Line 203, check the font size, not consistent. 

Line 189 increased by xxx, respectively. 

English needs to be checked by native speakers

Author Response

In response to comments from Reviewer 3:

Thank you for your approve. We agree with all comments from the reviewer and have revised carefully, as requested.

We have revised References format very carefully according to journal instructions and guidelines. Thank you for noticing.

Line 21, We have included the full name of RWC, relative water content, Line 158, as requested. Thank you for your suggestion.

Line 35-36. We have changed the Refs, as requested. Thank you for your suggestion.

Line 46-47. We have added the Refs at line 389-393, as requested. Thank you for your suggestion.

Fig. 1 We have revised figure 1, changed to RWC(%) on Y axis and deleted "stopped" on X axis. Thank you for noticing.

Line 189-Line 203, We have checked the font size here and revised it consistent. Thank you for noticing.

Line 189  We have revised it to "increased by xxx, respectively." here, also checked and revised all through the manuscript.  Thank you for noticing.

We have double-checked and revised the English language all through the manuscript by a colleague fluent in English writing. Thank you.